# Voltage-Dependent Anion Channel 1 Expression in Oral Malignant and Premalignant Lesions

**DOI:** 10.3390/diagnostics13071225

**Published:** 2023-03-24

**Authors:** Irit Allon, Jacob Pettesh, Alejandro Livoff, Mark Schlapobersky, Oded Nahlieli, Eli Michaeli

**Affiliations:** 1Institute of Pathology, Barzilai University Medical Center, Ashkelon 7830604, Israel; 2School of Health Sciences, The Ben-Gurion University of the Negev, Beer-Sheba 84105, Israel; 3Oral Medicine Unit, Barzilai University Medical Center, Ashkelon 7830604, Israel; 4Department of Oral & Maxillofacial Surgery, Barzilai University Medical Center, Ashkelon 7830604, Israel

**Keywords:** VDAC1, squamous cell carcinoma, oral cavity, oral epithelial dysplasia, immunohistochemistry, mitochondria

## Abstract

Background: The voltage-dependent anion channel 1 protein (VDAC1) plays a role in cellular metabolism and survival. It was found to be down or upregulated (overexpressed) in different malignancies but it was never studied in application to oral lesions. The purpose of this study was to retrospectively evaluate the expression of VDAC1 in biopsies of oral premalignant, malignant, and malignancy-neutral lesions and to examine the possible correlations to their clinicopathological parameters. Materials and methods: 103 biopsies including 49 oral squamous cell carcinoma, 33 epithelial dysplasia, and 21 fibrous hyperplasia samples were immunohistochemically stained with anti-VDAC1 antibodies for semi-quantitative evaluation. The antibody detection was performed with 3,3′-diaminobenzidine (DAB). The clinicopathological information was examined for possible correlations with VDAC1. Results: VDAC1 expression was lower in oral squamous cell carcinoma 0.63 ± 0.40 and in oral epithelial dysplasia 0.61 ± 0.36 biopsies compared to fibrous hyperplasia biopsies 1.45 ± 0.28 (*p* < 0.01 for both; Kruskal–Wallis test). Conclusion: Oral squamous cell carcinoma and epithelial dysplasia tissues demonstrated decreased VDAC1 protein expression if compared to fibrous hyperplasia samples, but were not different from each other, suggesting that the involvement of VDAC1 in oral carcinogenesis is an early stage event, regulating cells to live or die.

## 1. Introduction

Head and neck malignant tumors account for more than 800,000 cases and 450,000 deaths annually, out of which oral squamous cell carcinoma (OSCC) is the most prevalent type that mainly arises from long-standing potentially malignant oral lesions [1,2,3]. Despite advanced surgical techniques and new therapeutic strategies, the mortality rate for OSCC remains high in most countries, with an overall 5-year survival rate below 50% [1]. Leukoplakia, erythroplakia, oral submucous fibrosis, and some other oral lesions may develop into cancer via hyperkeratosis and hyperplasia and various degrees of oral epithelial dysplasia (OED), to carcinoma in situ, and eventually to invasive OSCC [4,5]. The early detection of OSCC and premalignant lesions that precede OSCC’s development is key to reducing mortality. Therefore, there is a need to introduce new markers that could predict this transformation of premalignant lesions into carcinoma. Such potential markers for OSCC progression may act as possible biological targets for future adjuvant or neo-adjuvant therapy.

The voltage-dependent anion channels (VDAC) are protein-regulated pores located at the outer membrane of the mitochondria, allowing the traffic of metabolites across the cytosol and mitochondria. Three isoforms are characterized in mammals as VDAC1, VDAC2, and VDAC3. Of them, the VDAC1 protein, being an anion channel, is the most widely expressed (5% of the total mitochondrial proteins). It is responsible for ions and nucleotide flux such as ATP/ADP, NADH/NAD, and calcium ions, supports glycolysis, functions as a platform for proteins, and prevents or initiates apoptosis [6,7,8,9,10,11,12,13,14]. This pro or anti-apoptotic VDAC1 activity regulates the cell’s life and death and is particularly relevant in cases of malignancies. The VDAC1′s involvement in cell death or survival indicates that this protein might be a prime regulator target for treating cancer.

The involvement of VDAC1 in oncologic processes was extensively studied during the last decade, but these studies produced various results. It appeared that numerous human cancer cell lines exhibited either elevated or decreased VDAC1 expression if compared to normal cell lines [15,16,17,18,19,20,21,22]. For specific examples, VDAC1 was overexpressed in breast cancer [23] but appeared to be low in endometrial cancer tissues [24]. VDAC1 expression in oral cancers and other oral conditions in general and OSCC, in particular, was not yet studied.

The authors aimed at analyzing VDAC1 expressions in malignant and premalignant oral lesions such as OSCC, as the most common oral malignancy, OED, and fibrous hyperplasia (FH). The second goal was to explore the association between possible changes in VDAC1 expressions and clinicopathological parameters of these lesions. In view that VDAC1 expression can be either elevated or decreased, the null hypothesis was that it remains unchanged. In the Discussion section, we also aimed at comparing VDAC1 OSCC activity with VDAC1 activities in cases of squamous cell carcinomas of other locations.

## 2. Materials and Methods

The study was approved by the Institutional Review Board for Human Subjects, number 0058-19-BRZ V3.0. No informed consent was needed since the study was carried out on archived material and patients were not involved.

### 2.1. Study Sample

During the current retrospective study, biopsies were retrieved from the archives of the Institute of Pathology at the Barzilai University Medical Center, which serves as a tertiary referral hospital, between the years 2016 and 2020 and revised for diagnosis confirmation and selection of representative sections. The authors selected the biopsies that were supported by full medical records. The study population was composed of all patients presenting for evaluation and management of various oral lesions. The cases with incomplete medical records, biopsies of poor quality, and other than OSCC, OED, and FH oral lesions were excluded from the study.

The authors reviewed 173 cases and 103 cases with adequate biopsies were selected for the study. The study sample consisted of three groups: 49 OSCC, 33 OED, and 21 biopsies with FH. According to the existing classifications [4,5,25], the OED group included low-grade and high-grade dysplasia. The low-grade dysplasia category was composed of biopsies indicating dysplastic changes (architectural and cytological) limited to the lower third of the epithelium thickness. The high-grade dysplasia category consisted of moderate to severe epithelial dysplasia biopsies, meaning dysplastic changes involving larger segments of epithelium thickness.

Clinical and pathological parameters were collected from the medical records and included demographic and clinical information such as age, gender, localization of the lesion, and TNM staging (AJCC 8th ed., 2018). The associated risk factors for OSCC were recorded and grouped as follows: Group 0—no associated risk factor, group 1—either smoking or alcohol consumption, group 2—post-transplant malignancy and/or immunosuppression therapy, and group 3—more than one risk factor (example: smoking and alcohol consumption). The cases of the OSCC group were further evaluated for histopathological risk assessment parameters according to Brandwein et al. [26]: worst pattern of invasion (WPOI) scored 1–5, accompanying inflammation scored 0, 1, or 3 and perineural invasion scored 0, 1, or 3. The overall risk assessment score was given on a scale of 0–9 and classified into low, intermediate, and high risk for local recurrence and overall survival probability.

### 2.2. Immunohistochemistry and Immunomorphometry

VDAC1 antibody was used to stain formalin-fixed paraffin-embedded human tissues of the three groups. Immunohistochemical staining was performed on positively charged slides (Leica Biosystems Richmond Inc., Richmond, IL, USA) incubated overnight in a humidified chamber at 60 °C degrees with primary VDAC1 antibodies (Rabbit anti-Voltage Dependent Anion Channel 1, 1:250, Darmstadt, Germany), and antibody detection was performed with 3,3′-diaminobenzidine (DAB) according to the manufacturer’s protocol. Briefly, following deparaffinization, the slides were warmed to 72 °C and cell conditioning was carried out. After that, the slides were warmed to 95 °C and incubated for 8 min. They were subsequently incubated for 20, 36, 52, and 64 min. The slides were warmed to 37 °C and incubated for another 4 min with primary antibody added for 32 min, ultra-washed, and counterstained. CellSens Dimension camera (version 1.8.1, Olympus, Shinjuku, Tokyo, Japan) was used to photograph the images in an X10 magnification. The intensity of the VDAC1 staining was measured with ImagePro Plus (version 6.0, Media Cybernetics, Rockville, MD, USA). According to the protocol, myocardium samples were used as positive controls due to their abundant mitochondria content. Semi-quantitative measurements were performed by a senior oral pathologist (IA) on all 103 slides as previously described [27]. The areas of interest within the relevant epithelial tissue samples in every slide were divided into 10 consecutive fields of analysis. The scoring was carried out on a scale of 0 to 3 separately for each of the 10 fields according to the staining intensity: fewer than 5% positive cells of the inspected field were given a score of 0, between 5 and 25% of the positive cells were given a score of 1, between 26% and 50% of the positive cells were given a score of 2 and a score of 3 was given for staining intensity above 50%. Then, a mean score was given for each case. Following this, a mean score for each of the study groups was given so these could be compared.

The expression of VDAC1 (the primary predictor variable) was associated with clinical variables such as age, gender, site of occurrence, TNM staging, and the associated risk factors for OSCC as covariates, and histopathological diagnosis as primary outcome.

### 2.3. Statistical Analysis

The data were analyzed by a biostatistician by RStudio software version 1.2.5001 under the following statistical assumptions: type 1 error of 5%; desired power of 90%; and a moderate effect size of the difference in the primary outcome between the three groups (*f* = 0.30). First, the differences between the groups OSCC, OED, and FH in demographic characteristics were analyzed. The continuous variable was examined with the Kruskal–Wallis test and discrete variables with the chi-square test. Post hoc analyses of the Kruskal–Wallis were conducted with the Mann–Whitney test, which was Bonferroni-corrected. Significance was set at *p* ≤ 0.025 and *r* ≥ 0.60 for correlations.

## 3. Results

### 3.1. Study Groups Demography

The demographic details are presented in Table 1. A significant age difference was found between the three groups, with FH cases with an average age of 47.95 ± 17.66 y in comparison to OED cases with 73.16 ± 12.45 y and OSCC cases with 67.90 ± 16.62 y (*p* < 0.01, for both, according to the Kruskal–Wallis test). No other statistically significant differences were detected between the groups. Table 2 presents the lesions according to the oral cavity subsites. The most common locations for OSCC and OED were the tongue, the lower lip, and the buccal mucosa.

Risk factors such as smoking or alcohol consumption were uncommon; the majority had no more than one risk factor (Table 2).

### 3.2. Staging Grading and Risk Assessment

In the OSCC group, twenty-three cases (47%) were T1, fourteen (29%) were T2, five (10%) were T3, four (8%) were T4A, and three (6%) were T4B. Most of these presented no neck involvement (67%), while eight (16%) patients were diagnosed with N1 disease, five (10%) with N2a, one (2%) with N2b (2 Nodes), and two (4%) patients with N3. Only two patients (4%) were M1 following metastasis to the lungs and spine. Risk score-wise, the majority were classified as a low risk 26 (53%) for local recurrence according to Brandwein et al. [26], while 23 (47%) were classified as intermediate and high-risk cases. In the OED group, 33% of the cases were high and 67% were low grades.

### 3.3. VDAC1 Expression

Descriptively, VDAC1 expression was confined to the cytoplasm and was abundant in the control myocardium (Figure 1). While some of the OSCCs positively expressed VDAC1, most of them showed decreased to absent VDAC1 expression compared to the FH samples (Figure 2A–C). The VDAC1 expression in the OED group was similar to the OSCC group and transition areas between normal appearing mucosa and OED/OSCC could be noted with VDAC1 decrease in the latter (Figure 2A).

Statistically, the mean expressions of VDAC1 levels in the OSCC and OED groups were 0.63 ± 0.40 and 0.61 ± 0.36, respectively, and were found to be significantly lower than the mean expression of VDAC1 in the FH group (1.45 ± 0.28); *p* < 0.01 for both (the Kruskal–Wallis test) (Figure 3A,B). At the cut-off level of 50 y, VDAC1 expression in younger participants was higher in comparison with older participants, rendering a negative correlation between age and VDAC1 levels (*r* = −0.30, chi-square test; *p* = 0.03, the Kruskal–Wallis test conducted with Mann–Whitney test). When aiming at correlating VDAC1 expression with histopathological risk assessment scores, VDAC1 was higher in the high and intermediate-risk assessment groups (0.91 ± 0.33) compared to the low-risk group 0.38 ± 0.18 (*p* = 0.02, the Kruskal–Wallis test).

No significant differences were found between VDAC1 levels and any of the demographic, clinical, or pathological parameters: gender (*p* = 0.55; correlation with gender r = 0.23), risk factors (OED and OSCC; *p* = 0.08, *p* = 0.03, respectively), tumor size (*p* = 0.41), nodes involvement (*p* = 0.38), metastasis (a group too small for statistical analysis), WPOI (*p* = 0.44), PNI (*p* = 0.23), accompanying inflammation (*p* = 0.77), and the three risk assessment groups (*p* = 0.09; correlation with risk assessment groups r = 0.36).

## 4. Discussion

In the current study, we aimed at characterizing the expression of VDAC1 protein in OSCC, OED, and FH tissues, i.e., in cancer, a premalignant lesion, and a reactive lesion with no propensity towards transformation. The results revealed lower VDAC1 expression in OSCC and OED compared to FH, indicating decreased expression in oral malignancy. Therefore, our null hypothesis was rejected. While significant differences in VDAC1 expression between OED and OSCC were not found, the molecular changes involved in VDAC1 expression might be seen as early events in carcinogenesis, such as a preliminary field cancerization-like effect. The average age differences between the study groups might have influenced VDAC1 expression as FH patients were younger than OED and OSCC patients. This finding could be responsible to some degree for the reduction in VDAC1 expression in malignant and premalignant lesions, although our literature review failed to find such a link.

In line with the present study, VDAC1 was found to be down-regulated in human esophageal carcinoma [15] and in melanoma (cell lines UACC903) [16]. Specifically for esophageal carcinoma, the gene expression profiles from patients with a family history of esophageal cancer were analyzed by cDNA microarray and VDAC1 was one of the calcium signaling pathways genes that were found to be down-regulated. The situation may be similar in the oral cavity, a site with both physical and system proximity to the esophagus.

However, VDAC1 is overexpressed (upregulated) in cervical carcinoma (Hela cell line) [28], adrenocortical carcinoma (Human SW13 and H295R) [17], myeloma [18], lung cancer [19], non-small cell lung carcinoma [20], human gastric carcinoma [21], and ovarian and endometrial carcinoma (OV-TRL12B, EN-TRL 67 T cell line) [22]. VDAC1 is capable of working in different directions, as it has a dual role as a regulator with opposite pathways. On one hand, it supports cancer cells’ bioenergy, metabolism, and anti-apoptotic activity and on the other hand, promotes mitochondria-mediated apoptosis. This variable behavior of VDAC1 in different cancers was recently summarized as “the intersection of cell metabolism, apoptosis, and diseases” [29]. Bioenergy and mitochondria-mediated apoptosis are the two main opposite pathways that are strongly associated with VDAC1 involvement and cell life and death (Figure 4).

By association with hexokinase (HK), VDAC1 favors both the production of energy, required by demanding cancer cells, and the protection against cell death, by decreasing the channel conductance for the passage of pro-apoptotic proteins to the cytosol. In contrast, increased VDAC1 expression is associated with its’ oligomerization, forming a larger pore that allows the release of pro-apoptotic proteins such as cytochrome c (Cyto c) and apoptosis-inducing factor (AIF) to the cytosol [9,10,11,12,13,14]. VDAC1 diameter is sufficient to allow the passage of nucleotides and small molecules but insufficient for a large folded protein such as pro-apoptotic proteins to pass. In response to an apoptotic stimulus, VDAC1 conformation changes to form an oligomer of six cylindrical monomers, which creates a channel wide enough to allow passage of pro-apoptotic proteins such as Cyto c and AIF. Hence, VDAC1 over-expression is associated with oligomerization and the release of pro-apoptotic proteins, leading to cell death [11]. Elevated calcium concentration within the mitochondria is another metabolic pathway associated with mitochondria-mediated apoptosis via Cyto c release. VDAC1, which lies at the junction between the cytoplasm and the mitochondria, functions as a pore for calcium passage from the *endoplasmic reticulum* to the mitochondria and, in addition, interacts with Bcl-2 family proteins which elevate the outer mitochondrial membrane permeability in response to stress [30,31]. Down-regulation of VDAC1 decreases calcium uptake by the mitochondria in cancer cell lines as in the human cervix and in the colon, and diminishes the activation of the mitochondrial intrinsic apoptosis pathway, consequently promoting cancer cell survival and tumor metastasis [31]. In principle, VDAC’s involvement in mitochondria-mediated cell death can be used for cancer treatment but this option was not yet fully materialized [32]. To date, the only positive results were achieved when VDAC1 silencing was applied to the treatment of mesothelioma [33].

### 4.1. VDAC1 Behavior in OSCC against Its Activities in Other Squamous Cell Carcinomas

The PubMed search for “VDAC1 cancer” presented 297 results to date. Of them, only nine mentioned squamous cell carcinoma. Our study is the tenth report on the subject. VDAC1′s activity was never studied for OED and FH cases before. Why VDAC1 expression in squamous cell carcinoma cases did not attract much attention is difficult to tell. Several solid tumors undergo total metabolic reprogramming in which glycolysis predominates, a phenomenon known as “the Warburg effect” [12,34]. The high glycolysis rate is mainly attributed to VDAC1 attachment to HK, considered the rate-limiting enzyme of this process [12,13,14]. HK attachment blocks VADC1 interactions with pro-apoptotic Bax proteins. It leads to preventing Cyto c release from the outer mitochondrial membrane and consequently decreases mitochondria-mediated apoptosis [11,34]. Malignancies that overexpress mitochondria-bound HK, such as colon, prostate, lymphoma, glioma, gastric adenomas, and breast cancers, demonstrate the important role that VDAC1-bound HK plays in cell bioenergy, growth rate, and survival of cancer cells [13,17,21,22].

This VDAC1–HK interaction was reported by Shoshan-Barmatz et al. [35] in general. For squamous cell carcinoma, the topic was researched for cervical carcinoma [36]. The authors of this study found that “the amount of hexokinases those bind to VDAC1 but not cell total hexokinases is decreased”. In addition to cervical carcinoma, VDAC1 expression was investigated only for lung, head and neck, and esophageal squamous cell carcinoma [15,36,37,38,39,40,41]. However, no other VDAC1-HK interactions in cases of squamous cell carcinoma were reported to date and this topic requires some further research.

The expression of VDAC1 in squamous cell carcinoma cases presents an additional problem. The most recent (2022) and detailed report of Wang et al. on VDAC1 “pan-cancer” activity presented an extensive list of tumor tissues in which VDAC1 was upregulated [42]. This list included “bladder urothelial carcinoma, breast invasive carcinoma, cholangiocarcinoma, colon adenocarcinoma, esophageal carcinoma, kidney chromophobe, kidney renal clear cell carcinoma, liver hepatocellular carcinoma, lung adenocarcinoma, lung squamous cell carcinoma, prostate adenocarcinoma, rectum adenocarcinoma, stomach adenocarcinoma, thyroid carcinoma, glioblastoma multiforme, and head and neck squamous cell carcinoma”. Within this list, the position of esophageal carcinoma is somewhat unclear because VDAC1, as mentioned above, was found to be down-regulated in human esophageal carcinoma [15]. In general, if all the above-mentioned squamous cell carcinoma reports are combined, we have the following picture. VDAC1 is overexpressed in cervical [37,38,40] and lung squamous cell carcinoma [41,43] being down-regulated in esophageal [15] and oral squamous cell carcinoma (our results). Its activity in head and neck cancers, in general, is not yet clear [39]. It was demonstrated that silencing VDAC1 expression inhibited cancer cell proliferation [19]. The question of how to approach VDAC1 which is already down-regulated, such as in OSCC cases, remains unclear. In addition to lining the oral cavity, stratified squamous epithelial cells are found in the skin and the vagina. Yet, no VDAC1-related reports were published on skin cancer (except skin cutaneous melanoma) and the cancer of the vagina, at least in PubMed-registered journals, to date. This topic also requires additional research, otherwise VDAC1 will remain a nonspecific marker.

Approaching this topic from a survival prognosis viewpoint, overexpression of VDAC1 is related to the poor overall survival of adrenocortical carcinoma, breast invasive carcinoma, cervical and endocervical cancers, glioblastoma multiforme, lung adenocarcinoma, pancreatic adenocarcinoma, and skin cutaneous melanoma cases, and low VDAC1 expression is associated with a poor overall survival prognosis of kidney renal clear cell carcinoma cases [42]. No VDAC1-related survival prognosis reports for cases of esophageal carcinoma and lung squamous cell carcinoma were published to date. As for our study, the overall risk assessment for survival probability did not indicate a poor overall survival prognosis for OSCC cases (*p* = 0.09).

### 4.2. VDAC1 Behavior in OED and Topic for Further Research

The similarity of VDAC1 expression in OSCC and OED cases deserves additional attention. While not cancer, OED was never studied for VDAC1 expression before our research. At the same time, it is generally accepted that OED is a premalignant lesion [3,44]. Recently (2021), Hankinson et al. reported that 2.6% of cases with mild OED, 4.1% of cases with moderate OED, and 29.2% of cases with severe OED progressed to OSCC at the dysplastic site and a small number developed a malignant lesion in other locations [45]. There is a wide field for further research in the future because bulky epithelial proliferation may be attributed to VDAC1 overexpression while epithelial atrophy may be a sign of apoptosis. However, while VDAC1 was found to be down-regulated in our analysis of OED cases, apoptosis may be associated with increased expression of VDAC-1 [46]. The first step of this research might be an investigation of VDAC1 behavior in cases of leukoplakia, erythroplakia, and oral keratosis. Of them, *proliferative verrucous leukoplakia* seems to be the main target for this research. The VDAC/HK2 modulator was asses for the treatment of actinic keratosis of the skin [47] but oral keratosis was overlooked.

Another direction for further research might be target therapy for OSCC that may involve VDAC1. Cisplatin, 5-azacytidine, and 5-fluorouracil, which are currently used as chemotherapy drugs against OSCC do not influence VDAC1 behavior [48,49,50]. Silencing VDAC1 is used for the treatment of mesothelioma and some other tumors [19,33] but this approach is inapplicable to OSCC because VDAC1′s expression is decreased already. Therefore, further research might concentrate on the implementation of nano-drug delivery systems and intelligent delivery systems to treat OSCC [51].

### 4.3. The Strength and Limitations of the Study

The study’s main strength is the first-time demonstration of VDAC1 expression in cases of various oral cavity lesions, such as oral squamous cell carcinoma, oral epithelial dysplasia, and fibrous hyperplasia, which represented malignant, premalignant, and malignancy-neutral oral lesions. For the limitations, this study was retrospective in design. We were unable to specify smoking and alcohol consumption habits because of the retrospective nature of our research. The study used the biopsies that were obtained from 2016 to 2020 with grading and other classifications available during this time period.

## 5. Conclusions

Oral squamous cell carcinoma and epithelial dysplasia tissues demonstrated decreased VDAC1 protein expression if compared to fibrous hyperplasia samples, but were not different from each other, suggesting that the involvement of VDAC1 in oral carcinogenesis is an early stage event, regulating cells to live or die. This decreased VDAC1 expression in OED and OSCC tissues is an early event and may be a component of the tumor cells’ mechanism for escaping apoptosis. Future studies on OED progression to OSCC should elucidate the role of VDAC1 on OSCC development.

## Figures and Tables

**Figure 1 diagnostics-13-01225-f001:**
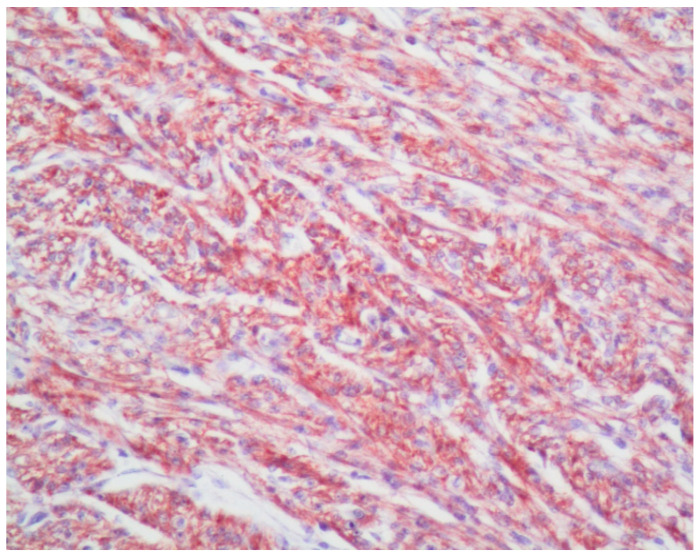
Diffuse cytoplasmic VDAC1 staining of the human myocardium served as the positive control (VDAC1 magnification ×100).

**Figure 2 diagnostics-13-01225-f002:**
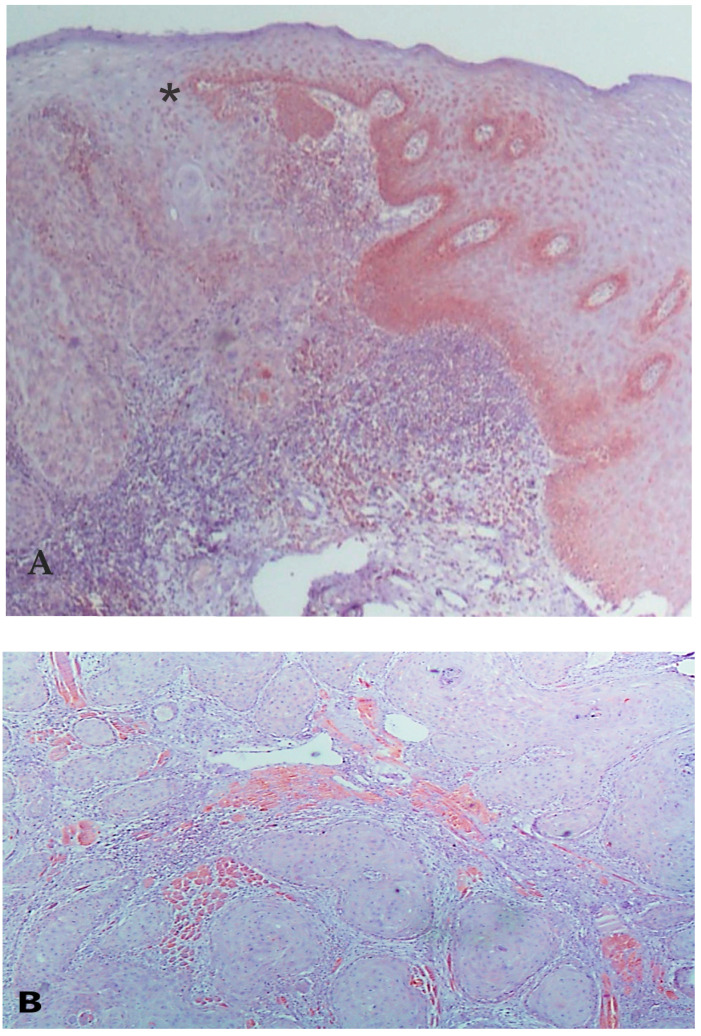
(**A**): A clear transition zone between normal-appearing mucosa with high VDAC1 expression and dysplastic/neoplastic mucosa with decreased VDAC1 expression (asterisk) (magnification ×100). The dysplastic grade is evaluated as “high grade”. (**B**): OSCC of the tongue showing decreased to absent VDAC1 staining of tumor islands invading the positively stained myocytes, a positive internal control (VDAC1 ×100, inset VDAC1 magnification ×200). (**C**): OSCC of the oral mucosa presenting decreased VDAC1 staining, while some inflammatory cells express VDAC1 (VDAC1; magnification ×400).

**Figure 3 diagnostics-13-01225-f003:**
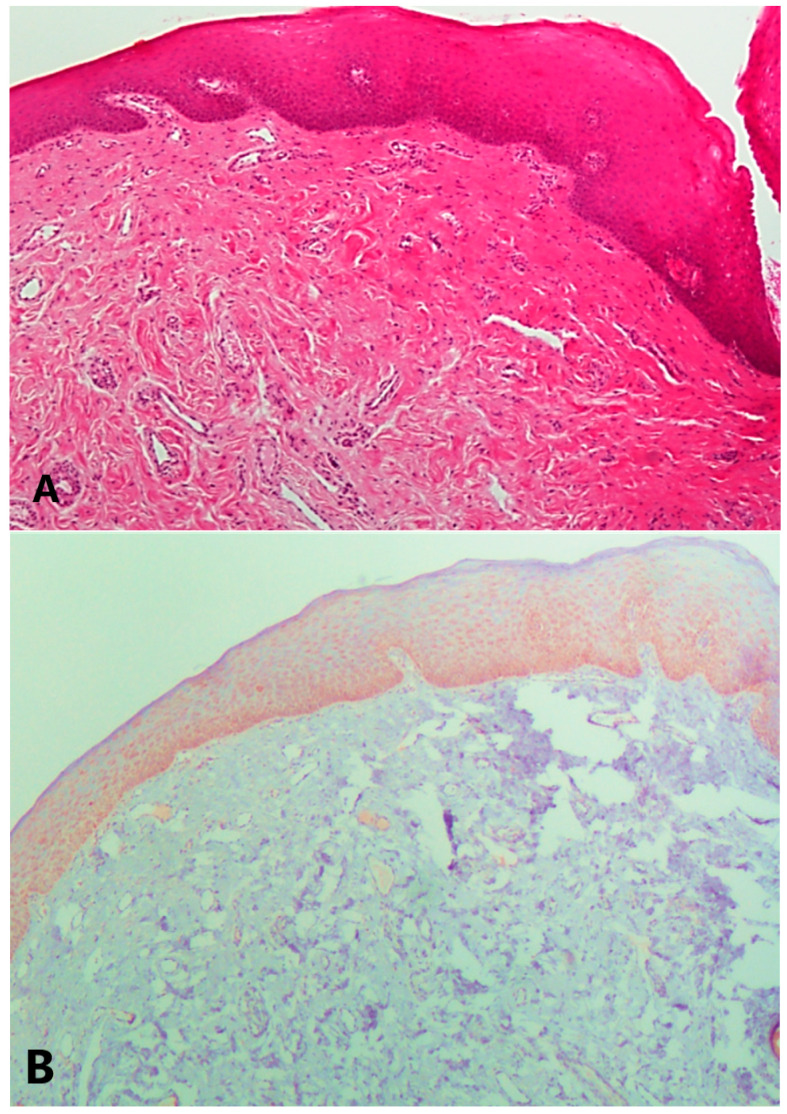
(**A**): Fibrous hyperplasia of the oral mucosa: Dense fibrous connective tissue lined by hyperkeratotic oral epithelium (H&E magnification ×100). (**B**): fibrous hyperplasia of the oral mucosa, Diffuse cytoplasmic VDAC1 staining in the epithelium. (VDAC1 magnification ×100).

**Figure 4 diagnostics-13-01225-f004:**
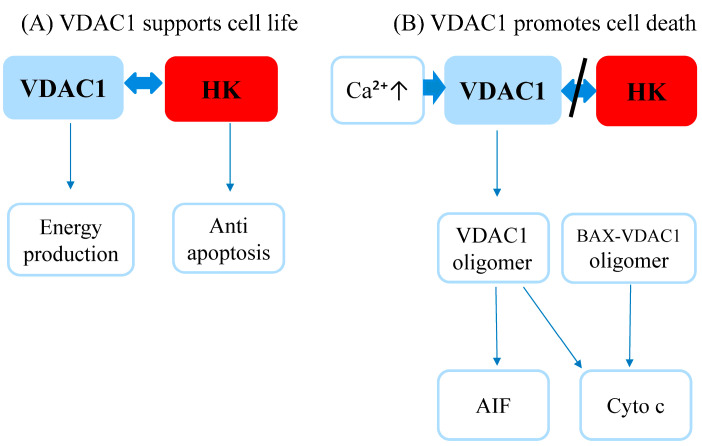
VDAC1 live or die flow chart: a schematic representation of VDAC1 involvement in cell survival (**A**) or death (**B**). (**A**) VDAC1 attaches HK, inducing cell energy production and protection from mitochondria-mediated apoptosis by blocking the interactions with pro-apoptotic BAX proteins. (**B**) Elevated Ca^2+^ levels and detachment of HK from VDAC1 promotes the oligomerization of VDAC1 to create large pores capable of passing apoptogenic proteins AIF and Cyto c. In addition, VDAC1 forms an oligomer complex with BAX proteins to create a pathway for Cyto c. Abbreviations: VDAC 1—voltage-dependent anion channel 1; HK—hexokinase; Cyto C—cytochrome c; AIF—apoptosis-inducing factor; Ca—calcium.

**Table 1 diagnostics-13-01225-t001:** Demographic details of the study groups.

	ⁱ OSCC	ⁱ OED	ⁱ FH
Mean age	67.9 ± 16.62	73.16 ± 12.45	47.94 ± 17.66
Age range	22–81	40–91	29–91
M:F ratio	1.7:1	1.5:1	1:1

ⁱ OSCC—oral squamous cell carcinoma; OED—oral epithelial dysplasia; FH—fibrous hyperplasia.

**Table 2 diagnostics-13-01225-t002:** Study groups sites of the lesions, risk factor data, staging, grading, and OED dysplasia severity.

	OSCC	OED	FH
Tongue	21 (43%)	9 (27%)	10 (48%)
Lower lip	7 (14%)	13 (39%)	-
Buccal mucosa	6 (12%)	6 (18%)	8 (38%)
Floor Of Mouth	5 (10%)	1 (3%)	-
Other sites ⁱ	10 (21%)	4 (13%)	3 (14%)
Risk factor group 0	20 (51.3%)	27 (51.3%)	20 (95.2%)
Risk factor group 1	19 (48.7%)	5 (15.2%)	1 (4.8%)
Risk factor group 2	7 (14.2%)	-	-
Risk factor group 3	3 (7.7%)	-	-
Staging:			
T 1	23 (47%)	-	-
T 2	14 (29%)	-	-
T 3	4 (10%)	-	-
T 4a	5 (8%)	-	-
T 4b	3 (6%)	-	-
N 0	33 (67%)	-	-
N 1	8 (16%)	-	-
N 2	5 (10%)	-	-
N 2b	1 (2%)	-	-
N 3	2 (4%)	-	-
M 0	47 (96%)	-	-
M 1	2 (4%)	-	-
Risk assessment:			
Low	26 (53%)	-	-
Intermediate	11 (22%)	-	-
High	12 (24%)	-	-
Dysplasia severity:			
Low grade	-	7 (21%)	-
High grade	-	11 (33%)	-
Solar cheilitis	-	12 (36%)	-
Atypia	-	3 (10%)	-

ⁱ Other sites with lesser than 5% distribution included: mandibular alveolar mucosa, mandibular vestibulum, chin, maxillary sinus, infraorbital, and palate.

## Data Availability

The research data is available by request to the corresponding author.

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
