# Peer review of "Voltage-Dependent Anion Channel 1 Expression in Oral Malignant and Premalignant Lesions"

_diagnostics, 2023, doi:10.3390/diagnostics13071225_

Round 1

Reviewer 1 Report

Dear Authors, 

I have revised your work entitled “Voltage Dependent Anion Channel 1 expression in oral malignant and premalignant lesions: Live or let die”. I congratulate with you for this interesting and well conducted study. Here are some suggestions to improve the manuscript. Please provide a point-by-point response, highlighting the corrections with a different color mark.

-       English should be improved along with the text.

-       Introduction: The null hypotheses of the study should be added at the end of the Introduction and rejected/accepted in the Discussion.

-       I suggest improving the aim of the study reported at the end of the introduction section.

-       Authors should improve the state of the art on OSCC, using more recent references, especially on the therapeutic target. At this regard, I suggest reading the following papers: DOI: 10.3390/cells10082127; DOI: 10.1007/s00432-022-04028-8.

-       I suggest making all the figures the same size (especially figure 1). 

-       Authors should add on the magnification scale into the figures. 

-       Replace figure 4 with one with higher resolution.

-       Please, remove typo errors along with the text. 

Author Response

“Diagnostics” Revision Reply letter

Reviewer 1

Dear Authors, 

I have revised your work entitled “Voltage Dependent Anion Channel 1 expression in oral malignant and premalignant lesions: Live or let die”. I congratulate with you for this interesting and well conducted study. Here are some suggestions to improve the manuscript. Please provide a point-by-point response, highlighting the corrections with a different color mark.

REPLY: We thank the Reviewer for his/her encouraging remarks and hope that our revisions will have done justice to his/her kind words. We highlighted all the changes in red as was suggested.

-       English should be improved along with the text.

REPLY: The text was re-edited.

-       Introduction: The null hypotheses of the study should be added at the end of the Introduction and rejected/accepted in the Discussion.

REPLY: We added in the Introduction:

“In view that VDAC1 expression can be either elevated or decreased, the null hypothesis was that it remains unchanged.”

We added in the Discussion:

“The results revealed lower VDAC1 expression in OSCC and OED compared to FH, indicating decreased expression in oral malignancy. Therefore, our null hypothesis was rejected.”

-       I suggest improving the aim of the study reported at the end of the introduction section.

REPLY: We reworded:

“The authors aimed to analyze VDAC1 expressions in malignant and premalignant oral lesions such as OSCC, as the most common oral malignancy, OED, and fibrous hyperplasia (FH). The second goal was to explore the association between possible changes in VDAC1 expressions and clinicopathological parameters of these lesions.”

-       Authors should improve the state of the art on OSCC, using more recent references, especially on the therapeutic target. At this regard, I suggest reading the following papers: DOI: 10.3390/cells10082127; DOI: 10.1007/s00432-022-04028-8.

REPLY: We added the following paragraph to the Discussion with four new references:

“Another direction for further research might be target therapy for OSCC that may involve VDAC1. Cisplatin, 5-azacytidine, and 5-fluorouracil, which are currently used as chemotherapy drugs against OSCC do not influence VDAC1 behavior [49-51]. Silencing VDAC1 is used for the treatment of mesothelioma and some other tumors [19, 33] but this approach is inapplicable to OSCC because VDAC1’s expression is decreased already. Therefore, further research might concentrate on the implementation of nano-drug delivery systems and intelligent delivery systems to treat OSCC [52].”

  1. Giorgini, E, Sabbatini, S, Rocchetti, R, et al. In vitro FTIR microspectroscopy analysis of primary oral squamous carcinoma cells treated with cisplatin and 5-fluorouracil: A new spectroscopic approach for studying the drug–cell interaction. Analyst 2018;143:3317–3326.
  2. Notarstefano, V, Sabbatini, S, Pro, C, et al. Exploiting fourier transform infrared and Raman microspectroscopies on cancer stem cells from oral squamous cells carcinoma: New evidence of acquired cisplatin chemoresistance. Analyst 2020;145:8038–8049.
  3. Notarstefano V, Belloni A, Sabbatini S, et al. Cytotoxic Effects of 5-Azacytidine on Primary Tumour Cells and Cancer Stem Cells from Oral Squamous Cell Carcinoma: An In Vitro FTIRM Analysis.Cells 2021;10(8):2127. doi: 10.3390/cells10082127.
  4. Li H, Zhang Y, Xu M, Yang D. Current trends of targeted therapy for oral squamous cell carcinoma. J Cancer Res Clin Oncol 2022;148(9):2169-2186. doi: 10.1007/s00432-022-04028-8.

-       I suggest making all the figures the same size (especially figure 1). 

-       Authors should add on the magnification scale into the figures. 

-       Replace figure 4 with one with higher resolution.

REPLY: Magnifications were included in the captions. All other issues will be taken care of at the stage of production. We have original TIFF files for all figures. During the submission stage, all figures were just embedded in the Word file of the text. Everything will be rearranged during converting this Word file into a page-by-page PDF file of the article. 

-       Please, remove typo errors along with the text. 

REPLY: The text was re-edited.

Reviewer 2 Report

The manuscript made by Allon I et al. It is related to study VDAC1 in different lesions including oral squamous cell carcinoma. They showed the association of expression by immunohistochemistry of VDAC1 with several variables and oral mucosa lesions. Despite of interesting study, I have comments that authors need to resolve.

Introduction it's interesting, but the redaction is confused; introduction needs be rewritten with a better explanation about variables (VDAC and OSCC) for example: "To date, no studies appeared in SCOPUS, ScienceDirect, and PubMed-registered journals regarding VDAC1 expression in oral cancers and other oral conditions in general and in OSCC in particular" these lines, that would it be confusing to readers.
"This study aimed..." Same to introduction the purpose of study need be rewritten in a better form, also authors were describing that did not have any hypothesis, when the hypothesis was written in these lines, please review.

Material and Methods
"Group 2 - post malignancy..." Why authors decided to include post malignancy (please review this term: "post malignancy") and immunosuppression therapy in the same variables, is there any reason to do it? Please explain.
Group 3 What was the difference between group 1 and group 2, both groups included same variables (smoking and alcohol)

OSCC group” Why was not included the WHO histopathological classification?  

2.2 immunohistochemistry
Please explain in detail or include reference related to DAB protocol.
I did not understand the difference between score 1 (0-25%) and score 2 (25%-50%) the 25% it is the same between group 1 and 2, please review and explain better.

"Therefore, VDAC1..." please explain better VDAC1 as a first variable and Clinical variables (Age,...) as a covariables. I suggest delete these lines and explain that expression of VDAC1 was associated with clinical variables and histopathological diagnosis.
Table 2. I suggest to authors instead of using risk factors, better, using the risk factors associated. for example,
                                     OSCC
smoking                          0%
consume of alcohol      0%
I recommend describing how many tobaccos were consumed per day, and classifying in light, moderate and heavy smoker, also classifying in past smoker, current smoker, smoker, non-smoker, and unknown.

Point 3.2
I suggest this point should be included in table 2.
That includes:
Histological grade (according to WHO classification, well, moderate and poor differentiated)
TNM and Clinical Stage (AJCC classification) and compare these variables with VDAC1 expression.
I suggest including Figure 1 into figure 2,
for example Figure 1, a) myocardium, positive control, b) A clear transition zone between....
The figure 2A is blurry and it is not possible to appreciate their characteristics; is it possible to include a better image?

Discussion
"The results revealed..." This suggestion is interesting since evaluate VDAC1 in malignant and non-malignant lesions, would it be possible make a correlation studies (Spearman or Pearson) between lesions and VDAC1 (OSCC vs VDAC1, OED vs VDAC1, FH vs VDAC1) and between lesions with expression of VDAC1 (OSCC VDAC expression vs OED VDAC1 expression, etc...) to highlighting the differences of VDAC1 expression and lesions?

2. "malignant melanoma" I suggest delete malignant.

4.1. VDAC behavior...
I think this point needs more detail, in which authors do hypothesis according to VDAC1 expression and histological grade, risk clinical data (age, smoking-alcoholism, TNM, Clinical Stage, histological grade etc...) In its current state, it seems more of an extension of introduction than a discussion properly. The authors need to evaluate all variables and statistical data related.

Overall Comments
This manuscript is interesting; however, authors need redesign study focusing mainly on malignant lesions; compare malignant vs benign lesions; associate better the variables with VDAC, review, and redesign the statistical studies taking account of all variables. The discussion needs to be related more to the results than hypothesis associated with the expression of VDAC1.

The title needs be change
English must be reviewed by an expert.

Author Response

Reviewer 2

The manuscript made by Allon I et al. It is related to study VDAC1 in different lesions including oral squamous cell carcinoma. They showed the association of expression by immunohistochemistry of VDAC1 with several variables and oral mucosa lesions. Despite of interesting study, I have comments that authors need to resolve.

Introduction it's interesting, but the redaction is confused; introduction needs be rewritten with a better explanation about variables (VDAC and OSCC) for example: "To date, no studies appeared in SCOPUS, ScienceDirect, and PubMed-registered journals regarding VDAC1 expression in oral cancers and other oral conditions in general and in OSCC in particular" these lines, that would it be confusing to readers.

REPLY: We ask the Reviewer to please accept our thanks for the analysis of our paper. We are grateful for his/her having so clearly pointed out a number of issues that needed to be addressed in order to enhance the clarification of the paper. We have done so for each citation as follows: The Introduction was shortened and arranged the following way:

Paragraph 1 – Importance of OSCC and premalignant oral lesions

Paragraph 2 – Description and importance of VDAC

Paragraph 3 - VDAC1’s expression can be either elevated or decreased. The sentence in question was reworded as: “VDAC1 expression in oral cancers and other oral conditions in general and OSCC, in particular, was not yet researched.”

Paragraph 4 – The aim of the study and null hypothesis

"This study aimed..." Same to introduction the purpose of study need be rewritten in a better form, also authors were describing that did not have any hypothesis, when the hypothesis was written in these lines, please review.

REPLY: We reworded:

“The authors aimed to analyze VDAC1 expressions in malignant and premalignant oral lesions such as OSCC, as the most common oral malignancy, OED, and fibrous hyperplasia (FH). The second goal was to explore the association between possible changes in VDAC1 expressions and clinicopathological parameters of these lesions. In view that VDAC1 expression can be either elevated or decreased, the null hypothesis was that it remains unchanged.”

Material and Methods
"Group 2 - post malignancy..." Why authors decided to include post malignancy (please review this term: "post malignancy") and immunosuppression therapy in the same variables, is there any reason to do it? Please explain.
Group 3 What was the difference between group 1 and group 2, both groups included same variables (smoking and alcohol)

 REPLY: For the associated risk factors for OSCC, we just followed general guidelines:

Huber MA, Tantiwongkosi B. Oral and oropharyngeal cancer. Med Clin North Am. 2014;98(6):1299–1321. doi: 10.1016/j.mcna.2014.08.005. 

Winn DM, Lee Y-CA, Hashibe M, Boffetta P. The INHANCE consortium: toward a better understanding of the causes and mechanisms of head and neck cancer. Oral Dis. 2015;21(6):685–693. doi: 10.1111/odi.12342.

The groups do not overlap, because smoking and alcohol are in group 1. For group 2 we clarified:

“group 2 – post-transplant malignancy and/or immunosuppression therapy” as these risk factors also were indicated.

“OSCC group” Why was not included the WHO histopathological classification?  
REPLY: We fully understand the Reviewer’s concern. We did not do this because if we will split each group into these subgroups, we will have a small number of cases in each subgroup and no statistical analysis will be possible. That is why we narrowed the results to OSCC's present/absent issue.

2.2 immunohistochemistry
Please explain in detail or include reference related to DAB protocol.
I did not understand the difference between score 1 (0-25%) and score 2 (25%-50%) the 25% it is the same between group 1 and 2, please review and explain better.

 REPLY: The manufacturer’s DAB protocol was presented:

“Briefly, following deparaffinization, the slides were warmed to 72℃ and cell conditioning was carried out. After that, the slides were warmed to 95℃ and incubated for 8 minutes. They were subsequently incubated for 20, 36, 52, and 64 minutes. The slides were warmed to 37℃ and incubated for another 4 minutes and with primary antibody added for 32 minutes, ultra-washed, and counterstained.”

We also changed 25% to 26%.

"Therefore, VDAC1..." please explain better VDAC1 as a first variable and Clinical variables (Age,...) as a covariables. I suggest delete these lines and explain that expression of VDAC1 was associated with clinical variables and histopathological diagnosis.

REPLY: We reworded:

“The expression of VDAC1 (the primary predictor variable) was associated with clinical variables such as age, gender, site of occurrence, TNM staging, and the associated risk factors for OSCC as covariates, and histopathological diagnosis as the primary outcome.”

Table 2. I suggest to authors instead of using risk factors, better, using the risk factors associated. for example,
                                     OSCC
smoking                          0%
consume of alcohol      0%
I recommend describing how many tobaccos were consumed per day, and classifying in light, moderate and heavy smoker, also classifying in past smoker, current smoker, smoker, non-smoker, and unknown.

REPLY: We fully understand the Reviewer’s concern. We were unable to specify smoking habits because of a retrospective nature of our research. The examined charts indicated “smoking” but further details were not usually presented.

We added to the limitations:

“We were unable to specify smoking and alcohol consumption habits because of the retrospective nature of our research.”

Point 3.2
I suggest this point should be included in table 2.
That includes:
Histological grade (according to WHO classification, well, moderate and poor differentiated)
TNM and Clinical Stage (AJCC classification) and compare these variables with VDAC1 expression.

REPLY: Table 2 was completely rearranged and we replaced it with the new one. We fully understand the Reviewer’s concern about comparisons. We did not do this because if we will split each group into these subgroups, we will have a small number of cases in each subgroup and no statistical analysis will be possible. It is seen from Table 2 that the number of cases in various subgroups may vary from 1 to 47 with intermediate 3, 7, or 11. It makes sound statistics impossible.  

I suggest including Figure 1 into figure 2,
for example Figure 1, a) myocardium, positive control, b) A clear transition zone between....
The figure 2A is blurry and it is not possible to appreciate their characteristics; is it possible to include a better image?

REPLY: All these issues will be taken care of at the stage of production. We have original TIFF files for all figures. During the submission stage, all figures were just embedded in the Word file of the text. Everything will be rearranged during converting this Word file into a page-by-page PDF file of the article. 

Discussion
"The results revealed..." This suggestion is interesting since evaluate VDAC1 in malignant and non-malignant lesions, would it be possible make a correlation studies (Spearman or Pearson) between lesions and VDAC1 (OSCC vs VDAC1, OED vs VDAC1, FH vs VDAC1) and between lesions with expression of VDAC1 (OSCC VDAC expression vs OED VDAC1 expression, etc...) to highlighting the differences of VDAC1 expression and lesions?

 REPLY: Yes, it could be done but the results will be in concord with the Kruskal-Wallis test that we applied. We used the Chi-square test for some correlations. For example, “At the cut-off level of 50 y, VDAC1 expression in younger participants was higher in comparison with older participants, rendering a negative correlation between age and VDAC1 levels (r = -.30, Chi-square test; p = .03, the Kruskal-Wallis test conducted with Mann-Whitney test).”

  1. "malignant melanoma" I suggest delete malignant.

REPLY: Deleted.

4.1. VDAC behavior...
I think this point needs more detail, in which authors do hypothesis according to VDAC1 expression and histological grade, risk clinical data (age, smoking-alcoholism, TNM, Clinical Stage, histological grade etc...) In its current state, it seems more of an extension of introduction than a discussion properly. The authors need to evaluate all variables and statistical data related.

 REPLY:  The Reviewer is absolutely right. It looks like an extension of the introduction. Our original submission contained 2,200 words. We just reported the novel data on oral lesions adding this data to VDAC1-OSCC data obtained from other locations. The editors of the journal, however, asked for a 4,000 words text. We expanded the Discussion section by introducing the “VDAC1 behavior in OSCC against its activities in other squamous cell carcinomas” subsection. Yet, adding statistical data will move us to meta-analysis while we just report the results of our research.

Overall Comments
This manuscript is interesting; however, authors need redesign study focusing mainly on malignant lesions; compare malignant vs benign lesions; associate better the variables with VDAC, review, and redesign the statistical studies taking account of all variables. The discussion needs to be related more to the results than hypothesis associated with the expression of VDAC1.

REPLY: We introduced numerous changes in the text suggested by three Reviewers. Our research was based on the biopsies available in our hospital data bank. We compared malignant (OSCC) and benign (FH) lesions and detected the VDAC1 expression differences. At the same time, we found no difference in VDAC1 expression between malignant and premalignant (OED) lesions. We, therefore, stressed the importance of studying OED in this connection.

The title needs be change

REPLY: The title was shortened to Voltage Dependent Anion Channel 1 expression in oral malignant and premalignant lesions.

English must be reviewed by an expert.

REPLY: The text was re-edited.

Reviewer 3 Report

Thank you very much for the opportunity of review this work. Since my point of view is very interesting know that, metabolic issues in oral cancer is one of the most recent discussion. However I have some suggestions

·      In introduction section, in the first paragraph line 6, it is very convenient to delimit Oral premalignant lesions.

·      In materials and methods section:

-       Authors never clarify that fibrous hyperplasia (FH) was taken as a control, even author didn´t take the immunoreactivity of the collagenous fibers. They use as a control the cover epithelium of those benign neoplasm.

-       I highly recommend update the references of grading dysplastic changes with next reference doi.org:10.1007/s12105-021-01402-9

-       Please specify the AJCC edition used to staging TNM system

-       Take smoking or alcohol as the same risk factor is a weakness of the study in this must discussed. Is known that the oral epithelial dysplasia and oral cancer occurrence in alcohol drinkers who are nonsmokers, the role of alcohol is crucial only when considered in conjunction with tobacco.

·      In results section:

-       Add the p value to table 1

-       Results of staging grading and risk assessment must summarize in a table too. I recommend join with table 2.

-       Figure 2A must mention the dysplastic grade

·      In discussion section:

-       In figure 4 all molecule mentioned in the image, its abbreviation must be defined in figure legend.

-       Section 4.1:

-  The ten studies must be cited next to

-  When authors say “…VDAC1´s activity was never studied for OED and FH cases before…”, why they excluded OSCC? The previous nine works were about oral squamous cell carcinoma?

-  Third paragraph: In the narrative and the intention of pointing out the new position of esophageal carcinoma, it seems irrelevant make the list of all the carcinomas included.

-  Authors must consider for their discussion to contrast, primarly with other studies that developed a similar methodology. I mean only with immunoexpression intensities results and other type of expression as evaluate the RNAm levels or western blot analysis between others, they should be in a separate section. Because throughout the discussion, it seem that gene and protein expression could be interchangeable.

-  Finally, authors never mention the possible factors involve for the low VDAC1 immunoexpression.

Author Response

Reviewer 3

Thank you very much for the opportunity of review this work. Since my point of view is very interesting know that, metabolic issues in oral cancer is one of the most recent discussion. However I have some suggestions

  • In introduction section, in the first paragraph line 6, it is very convenient to delimit Oral premalignant lesions.

REPLY: We thank the Reviewer for his/her remarks and hope that our revisions will have done justice to his/her kind words. Following the Reviewer’s advice, we made several clarifications and amendments to the text.

We shortened this section to:

Leukoplakia, erythroplakia, oral submucous fibrosis, and some other oral lesions may develop into cancer via hyperkeratosis and hyperplasia and various degrees of oral epithelial dysplasia (OED), to carcinoma in situ, and eventually to invasive OSCC [4, 5].

This phrase is important to explain why we added OED and hyperplasia to our research on OSCC.

  • In materials and methods section:

-       Authors never clarify that fibrous hyperplasia (FH) was taken as a control, even author didn´t take the immunoreactivity of the collagenous fibers. They use as a control the cover epithelium of those benign neoplasm.

REPLY: Yes, we did not indicate FH as a control because a “control group” is usually healthy cases, but we did not have a significant number of healthy tissue biopsies in our database to present them as a clear control group.

-       I highly recommend update the references of grading dysplastic changes with next reference doi.org:10.1007/s12105-021-01402-9

REPLY: The study used the biopsies that were obtained from 2016 to 2020. We indicated this. The reports in the charts and from pathologists used the older grading. We added to the Limitations:

The study used the biopsies that were obtained from 2016 to 2020 with grading and other classifications available during this time period.

-       Please specify the AJCC edition used to staging TNM system

REPLY: We added: (AJCC 8th ed., 2018).

-       Take smoking or alcohol as the same risk factor is a weakness of the study in this must discussed. Is known that the oral epithelial dysplasia and oral cancer occurrence in alcohol drinkers who are nonsmokers, the role of alcohol is crucial only when considered in conjunction with tobacco.

REPLY: For the associated risk factors for OSCC, we just try to follow general guidelines:

Huber MA, Tantiwongkosi B. Oral and oropharyngeal cancer. Med Clin North Am. 2014;98(6):1299–1321. doi: 10.1016/j.mcna.2014.08.005. 

Winn DM, Lee Y-CA, Hashibe M, Boffetta P. The INHANCE consortium: toward a better understanding of the causes and mechanisms of head and neck cancer. Oral Dis. 2015;21(6):685–693. doi: 10.1111/odi.12342.

We were unable to specify smoking habits because of the retrospective nature of our research. The examined charts indicated “smoking” but further details were not usually presented.

We added to the limitations:

“We were unable to specify smoking and alcohol consumption habits because of the retrospective nature of our research.”

  • In results section:

-       Add the p value to table 1

REPLY: The P-values were not added to Table 1 because of two reasons: three groups were compared and all P-values except one were statistically insignificant. We, therefore, introduced the only statistically significant value in the text:

“A significant age difference was found between the three groups, with FH cases with an average age of 47.95 ± 17.66 y in comparison to OED cases with 73.16 ± 12.45 y and OSCC cases with 67.90 ± 16.62 y (p < .01, for both, the Kruskal-Wallis test).”

We also added:

No other statistically significant differences were detected between the groups.

-       Results of staging grading and risk assessment must summarize in a table too. I recommend join with table 2.

REPLY: Table 2 was fully redesigned.  

-       Figure 2A must mention the dysplastic grade

REPLY: We added to the legend: “The dysplastic grade is evaluated as “high”.

  • In discussion section:

-       In figure 4 all molecule mentioned in the image, its abbreviation must be defined in figure legend.

REPLY: We added to the legend:

Abbreviations: VDAC 1 - voltage-dependent anion channel 1; HK - Hexokinase; Cyto C - cytochrome c; AIF - Apoptosis-inducing factor; Ca – calcium.

-       Section 4.1:

-  The ten studies must be cited next to

-  When authors say “…VDAC1´s activity was never studied for OED and FH cases before…”, why they excluded OSCC? The previous nine works were about oral squamous cell carcinoma?

REPLY: We mentioned this already in the Introduction:

VDAC1 expression in oral cancers and other oral conditions in general and OSCC, in particular, was not yet researched.

-  Third paragraph: In the narrative and the intention of pointing out the new position of esophageal carcinoma, it seems irrelevant make the list of all the carcinomas included.

REPLY: The reason behind this is that we pointed out that in most of the carcinoma cases (presenting indeed this long list) VDAC1 is overexpressed (upregulated) while it appeared to be down-regulated in our OSCC cases.

-  Authors must consider for their discussion to contrast, primarly with other studies that developed a similar methodology. I mean only with immunoexpression intensities results and other type of expression as evaluate the RNAm levels or western blot analysis between others, they should be in a separate section. Because throughout the discussion, it seem that gene and protein expression could be interchangeable.

REPLY: Our research was based on the biopsies available in our hospital data bank. We compared malignant (OSCC) and benign (FH) lesions and detected the VDAC1 expression differences. At the same time, we found no difference in VDAC1 expression between malignant and premalignant (OED) lesions. We, therefore, stressed the importance of studying OED in this connection. No genetics methodology was involved and we cannot speculate on this issue.

-  Finally, authors never mention the possible factors involve for the low VDAC1 immunoexpression.

REPLY: Our research was based on the analysis of available biopsies. We detected the low VDAC1 expression but to answer the question “why”, some other type of research will be needed.

Round 2

Reviewer 2 Report

The manuscript improved, authors made a great effort to do it. Despite of the manuscript was improved, English need be reviewed by an expert, please provide a certificate of review.

Author Response

The English editing was performed with track changes this time 

Reviewer 3 Report

Thank you very much to attend the comments.

Author Response

again, thank you for your review!